# Remediation with Semicoke-Preparation, Characterization, and Adsorption Application

**DOI:** 10.3390/ma13194334

**Published:** 2020-09-29

**Authors:** George Lartey-Young, Limin Ma

**Affiliations:** 1College of Environmental Science and Engineering, Tongji University, 1239, Siping Road, Shanghai 200092, China; 1990030@tongji.edu.cn; 2Shanghai Institute of Pollution Control and Ecological Security, Shanghai 200092, China

**Keywords:** adsorption, activation, contaminant, regeneration, semi-coke

## Abstract

Development of low-cost contaminant sorbents from industrial waste is now an essential aspect of the circular economy since their disposal continues to threaten ecological integrity. Semicoke (SC), a by-product generated in large quantities and described as solid waste from gasification of low-rank coal (LRC), is gaining popularity in line with its reuse capacity in the energy industry but is less explored as a contaminant adsorbent despite its physical and elemental carbon properties. This paper summarizes recent information on SC, sources and production, adsorption mechanism of polluting contaminants, and summarizes regeneration methods capable of yielding sustainability for the material reuse.

## 1. Introduction

In recent years, much attention has been directed to treating and restoration of contaminated land and water bodies as a requirement for future land uses increases [1]. Several technologies have been explored for a variety of contaminants comprising in-situ chemical oxidation (ISCO), pump and treat, bioremediation, solidification, stabilization, and electrokinetics, among others [2]. Due to the nature of contaminant formation, specific remediation technologies, or sometimes a combination of two or more technologies, may be employed. Adsorption has become accepted as a cost-wise and efficient technology for the treatment of contaminated drinking water, wastewater, and soil [3]. An important aspect of adsorption technology is the application of effective and sustainable adsorbent (sorbent) material. Some commercial adsorbents include activated carbons, amorphous ferric oxyhydroxide (AFO), biomass waste, zeolites, and ion exchange resins [4]. Early classification of these adsorbents based on their raw material sources generated three categories; carbon, mineral, and other adsorbents [5]. The term ‘green adsorbents’ was later introduced to describe adsorbents obtained solely from forest products [6]. A narrower and simplistic classification into two groups as either conventional and non-conventional adsorbents was later reported [7]. Much recent and broader classification according to feedstock, by-products generated, and modifications, into five groups, are shown in Table 1 [1]. Among the conventional adsorbents, activated carbons ^(+++)^, zeolites ^(++)^, silica gel ^(+)^, and activated alumina^(-)^ have been found to be successful for commercial applications in order of magnitude listed [8], and a large body of literature exists on their contaminant adsorptive capacities. Nevertheless, the high cost of preparation, frequent requirement for regeneration, and the replacement of fresh materials during application limit their sustainability in the contaminant remediation process [4]. Scientific inquiry for identifying what is described as ‘low cost’ but high-performance adsorbents, based on natural bio-physicochemical properties or by engineered modification processes, is pioneering [9]. The term ‘low cost’ is discussed in association with biological and synthetic waste ‘solid waste’ or by-products from manufactured and industrial operations with the capacity for reuse as sorbents [10]. A very good ‘low cost’ adsorbent is proposed to possess significant carbonaceous structural formations, being capable of yielding an estimated contaminant removal rate within few hours to approximately eight hours daily, comparable to other carbon-based adsorbents [4]. Above all, a ‘low cost’ adsorbent should also be readily available and accessible.

Advancement in industrial modernization has triggered an ever-increasing demand for energy with the reliance and use of coal resources continuously growing to support such needs. The direct combustion of coal to produce energy is typically not sufficient and tends to generate some prominent environmental problems in its conversion process [11]. Therefore, diversification and transformative utilization of coal resources and end products can serve as an important means of achieving sustainability in resource use. Globally, low-rank coal (LRC) is abundant, making it low priced, and constitutes a significant component of energy-chemical feedstock [12]. A major by-product of the gasification, liquefaction, and pyrolysis of LRC is semi-coke (SC) [13]. The massive amounts of SC produced during the coal transformative process have resulted in it being described as a ‘solid waste’ with its disposal becoming an intractable problem [14]. Large amounts of SC are reported to be disposed of in open dumps, potentially threatening surface and groundwater pollution through leaching of polycyclic aromatic compounds and heavy metals [15]. Unlike fly-ash, which has found some significant post uses as filter material [16] and as an improvement agent for soil fertility [17], there is a paucity of literature regarding the application of SC, with perhaps its most important cited post usages for re-gasification in its “lump form” whilst the “powder form” yet remains an environmental threat [18].

Semicoke (SC) is generally characterized by lower moisture, ultralow volatile contents, low adhesiveness, and ultrafine particle size (<100 µm) with a typical block structure of 10–60 mm [19]. The carbon content alternates as low or high depending on its derivative source [20]. Its major composition comprises total soluble solids, carbonates, sulfate, bicarbonates, organic contaminants, and trace elements [21]. Following these characteristics, SC can be put into category 1 under Table 1. Since SC contains a large quantity of unburned carbon (30–70%), its re-utilization for environmental remediation can present a sustainable means of reducing its impact rather than disposal [13]. The potential transformation to high-value SC sorbent material will not only improve environmental gains, but have economic viability [22]. Recent advances in the transformation of SC have sought to produce a type of slurry fuel, semi-coke water slurry (SCWS) as potential clean material that can replace oil usage in the power combustion industry [23]. A similar product, described as lignite water slurry (LWS), was reported by [24].

This review provides a summary of the recent information on SC contaminant remediation application through a discussion of its nature (sources, types, production, physical, chemical properties, engineered modifications), and integration of SC application to general contaminant adsorption.

## 2. Materials and Methods

To obtain recent information and evidence on SC adsorption research, the search databases Web of Science and Scopus were explored with search items by combining ‘semi-coke’ as a constant word with other key terminologies, adsorption, removal, organic pollutants, heavy metals, and regeneration. The logic operator “AND” was used to refine all searches. A total of 303 records were retrieved. All records were imported to Endnote X8 and securitized for duplication before further selection analysis. By applying a designed qualitative spreadsheet, relevant articles were selected per title and abstract screening via the following criterion: (a) Articles that had titles bearing ‘semicoke adsorption/removal’ and (b) abstracts that contained keywords and information on ‘semicoke adsorption, utilization, pollutant removal’. A total of 46 articles were selected as recent adsorption-based research on SC within the year limits of (2010–2020) per the selection criterion and included in this review. Outcomes of the search result are shown in Figure 1, illustrating the paucity of adsorption research data in the study area. The designation “Other publication” comprised articles retrieved but related to other aspects of SC application e.g., re-gasification. Appendix A shows overview of search strategy.

## 3. Source, Type, and Production

### 3.1. Source and Type

The types of SC can be connected to four classes of LRCs in literature. By classification, LRCs can be grouped per parameters of the American Society of Testing and Materials (ASTM) and the International Standards Organization (ISO) [25]. In terms of originating sources, LRCs are reported to occur generally from Portugal through Thailand, Canada, Chile, China, and Antarctica with recent estimations representing approximately 55% of entire coal resources (Birkenmajer et al.) [25]. Depending on the source, SC tends to exhibit varying features that make its application and utilization unique. Lignite, or so-called ‘brown coal’, is by far the most described and reported source of SC production due to its characteristic reactive features [26]. It has its naturally oxidized form occurring as leonardite [27]. Other described sources include oil shale during retorting processing under high temperatures that have been found [28]. Oil sands or tar sands have also been investigated as sources for SC production [29]. Other bituminous and sub-bituminous raw coal and their naturally weathered form, humalite, have been described to produce SC [30]. Anthracite called ‘hard coal’ is a noted relative source but rarely reported [31]. Table 2 shows some specific features of the most-reported SC types. Amongst the three SC types indicated, lignite possesses highly described features, although it generally has a small surface area and hence may not, for instance, support physical adsorption mechanism. Bituminous coal, ‘soft coal’, SC, on the other hand possesses lower viscosities, hence easily undergoes shearing allowing the migration of substances into its molecular structure. Its adsorption effects are likely to be quicker and faster. In terms of surface volatile organic compounds, oil shale SC is higher in PAHs comparable to lignite and Bituminous coal SC. Figure 2 and Figure 3 show the classification of SC based on various feedstock and a generic flow chart of SC production factors, respectively.

### 3.2. Preparation

SC from any type of feedstock is produced under high heating rates and varying temperature regimes by pyrolysis, which is the degradation of a material in the absence of oxygen [38]. The pyrolysis process can be sub-classed into conventional pyrolysis, fast pyrolysis, flash pyrolysis, and low pyrolysis [39]. However, pyrolysis by fast and low heating rates is mostly reported on for SC production [40]. The formative process is characterized by first drying and desorption at temperatures >300 °C, followed by primary decomposition and depolymerization with the formation of SC and intermediates products of tar and gases [41]. Compared to slow pyrolysis, fast pyrolysis tends to decrease tar formation yields while gases such as Carbon monoxide (CO) volume fractions increase and a corresponding decrease in Carbon dioxide (CO_2_) in the process [12]. The intermediate tar formation results in low adhesiveness which limits SC’s adsorption efficiencies [42].

Figure 4 illustrates a generic flow process of SC formation from a typical LRC. Due to its macromolecular structure, relatively high heating (pyrolysis) is required to breakdown the networked matrix creating significant pores of varying sizes for intraparticle dispersion to take place. Low pyrolysis events may occur, however might affect its yield. Most of the surface and internal components of the SC are evaporated during initial pyrolysis events. The heating process further causes a transformation in aromatic and aliphatic structures at high temperatures resulting in significant losses and decomposition (e.g., from four rings to two rings) of aromatic structures, affecting yield [26]. However, this can be maintained at low-temperature pyrolysis events. These conversions are similar under the mentioned SC processing methods with slight differences occurring during the initial handling of the SC and heating periods.

Amongst the noted sources of SC, lignite and oil shale are difficult to ignite due to their low organic and mineral matter content [43]. Since SC is described as an environmental problem, the appropriate production methods to reduce its footprint is constantly under investigation. Pyrolysis time, coal particle size, and environmental pressure are classified as some essential ingredients to produce high yielding SC [11]. For instance, high-pressure impregnation and heat treatment methods have proven to yield good-performing desulfurization SC adsorbents [34]. The use of fluidized bed reactors [18], fixed bed reactors [44], and drop tube reactors [45] have recently been reported as reducing the amount of SC produced during gasification processes. A description of oil shale SC production methods including the Galoter retorting process, the ATP process, the Brazilz petrosix, Kivitier, and Fushun type retorts have been provided [46]. Although these SC traditional pyrolysis production methods are widely used, there are noted limitations such as slow heating velocities, hence requiring a longer time to initiate combustion and generate uneven heating, which affects the SC yield in terms of structure formation, which leads to lower efficiencies [47]. The ecologically clean ‘Termokoks’ process of producing SC from lignite was demonstrated by [48].

Microwave-assisted pyrolysis (MWAP) was recently described as an efficient and cleaner means of producing SC with referred frequencies in the ranges 915 to 2450 MHz [24]. Despite a cleaner process, it is often limited due to the dielectric properties of SC [49]. Hence, SC produced by MWAP may have low adsorptive capacities due to poor surface charge formations. A thorough comparison of traditional/conventional pyrolysis and MWAP was reported by [49] and the difference in pyrolysis product formation and composition was mainly related to differential heating mechanisms as the former displays an endothermic reaction process while the latter exhibits an exothermic process. A type of ultrasonic irradiation method of preparing SC. which can increase the dispersion of active components and increase surface area, was reported [50].

### 3.3. Characterization

Both chemical and physical characterization methods have been employed to describe SC. Typical of any ‘black carbonaceous’ formation, carbonization causes many changes including loss of functional groups, ordering of the carbon microstructure to become more graphitic, and a potential decrease of the inorganic matter catalytic role [51]. Characterization, therefore, enables identification and understanding of physicochemical transformations of the feedstock material after carbonization such as the establishment of high hydrogen-carbon (H/C) or oxygen-carbon (O/C) ratios. which determine their aromatic growth and maturation [28]. An understanding of its transformative process is therefore important t as it provides a theoretical understanding for its re-utilization [26]. The formations and assemblages of SC can be studied by observing, surface morphologies, crystalline structural formations, and diffraction analysis. The surface behavior and constituent composition are popularly determined by infrared spectroscopy, which can analyze its aromaticity growth [34]. Specific surface area (SSA) is conducted popularly and most reported on according to Brauner–Emmet–Taller (BET) method while a scanning electron microscopy analysis (SEM) is performed for surface morphologies.

A significant feature of SC characterization is the formation of micro, meso, and macropores for which pyrolysis temperature plays an important factor [41]. The temperature in the ranges or ≥ 1000 °C is reported as most efficient to achieve significant pore formation and these pore structures tend to act as the host of chemical reactions [18]. The linearity of temperature rise to structural alteration of SC was therefore confirmed by [15]. At high formation temperature, an opening of porous SC structures, is expected to facilitate adsorption and other internal exchange processes [52]. Longer pyrolysis duration tends to promote the complete removal of volatiles and results in more developed SC pores [11]. However, higher pyrolysis temperature may cause a severe chemical reaction and more pronounced damage to the carbon matrix thereby collapsing pore formations and disrupting any further intra-wall processes. SC micropores have a greater influence on high processes within the SC structures than macropores formations [53]. Pore structural formation further influences SC surface area activities [54]. In their study, [42] confirmed that SC surfaces can be finer than general porous carbon (PCs) as SC is compact and exhibits a non-porous structure unlike PCs with a large amount of pores spaces that may favor adsorbate diffusion. Table 3 shows some effects of varying temperature, modification methods, and agents on SC enhanced surface area (SSA) and pore volume for adsorption. The carbon and ash content are observed to increase proportionally with temperature rise while volatile matter on the SC surface decreases [11]. The organic matter (OM) content of oil shale SC has reported infractions of approximately 1.7–17.5% at pyrolysis of ≥500 °C comprising aromatic hydrocarbons, nitrogenous organic matter, hydrocarbons, ketones, and alcohols [46]. However, the majority of OM is lost during the pyrolysis process and this may contribute to its low adsorptive capacity in the natural state [15]. Recent reports have identified metallic fraction constituents such as SiO_2_, CaO, and K_2_O within SC which enhance support for needed high surface area and adsorption capacity [55]. A large quantity of the oxygen-containing functional group on SC’s surface makes it easy to be modified and used as a kind of adsorbent or catalyst [56]. However, the H and O concentrations that form hydroxyl groups tend to decrease with a rise in temperatures therefore only partially preserved, impacting the H/C atom ratios [57].

The elemental characteristics of SC are traditionally described by proximate and ultimate analysis as shown in Table 4 The proximate analysis measures the amount of moisture, ash, volatile matter, and fixed carbon while ultimate analysis measures the amount of carbon (C), hydrogen (H), nitrogen (N), sulphur (S), and oxygen (O) content. Fourier Transform Infrared spectroscopy (FTIR) coupled with Raman spectroscopy has been cited as the most-described method for elemental analysis [56]. Similar to its physical formation, at appropriate temperatures, SC obtains enhanced elemental properties which may enhance its adsorptive capacity.

### 3.4. Modification and Activation

The presence of large oxygen-containing functional groups on the SC surface enables it to undergo physical and chemical modification to enhance its capacity as a sorbent. According to [42], these may involve physical activation processes through CO_2_ or steam application at temperatures ≥ 700 °C. Steam activation is considered more efficient due to faster reaction kinetics between carbon ad steam [67].

Generally, chemical activation is conducted by adding constituents including KOH, NaOH, H_3_PO_4,_ K_2_CO_2_, and NaCl_2_. Through the introduction of surface activeness, the intrinsic properties of SC are ignited to interact between its carbon atoms [68]. Comparatively, chemical activation is much more useful as it promotes higher surface areas and controlled micropore distribution formation [69] while steam activation is considered significant than CO_2_ activation. KOH is indicated as an efficient activating agent, unlike HNO_3_, which may destroy SC pore structure and decrease surface areas [70]. The common methods of grinding and impregnation are possible routes to attain such modification, however since the former is likely to yield un-uniform mixing, the latter is most preferred [71]. Some researchers have reported on mixing SC with fly ash, which can alter the specific surface area and pore structure, implying that modification of SC with other chemical inherent carbon-based materials can enhance its efficiency of raw SC [26]. An elaborate description of hard and soft templating methods is provided by [72] but there is no evidence of application to SC. By soft templating, uniform SC structures can be produced with ordered micro and mesopore formation. Soft templating methods are described by [73]. The addition of oxidizing reagents such as H_2_O_2_, O_3_, and KMO_4_ has been explored for activating other carbon materials (e.g., biochar). It was found that by such modification, in this process, the carbon material obtained high hydrophilicity through increased acidic functional group formation unlike that observed during a typical thermal treatment process [74].

Heteroatom doping, a type of chemical activation, has recently been reported as an innovative means to enhance the functionality of carbonaceous materials, although there is no discoverable literature relating to SC. Doping by heteroatoms including (N, P, S, B, O, etc.) has been applied directly as precursors of activation [71]. Among heteroatoms, N has been widely reported on for doping carbonaceous materials as it enhances basicity, surface polarity, and modifies electronic structures [75]. They generate positively and negatively charged groups that are effective in adsorbing reactants or their intermediate daughter products, product desorption, and promote the overall conversion of reactants into desired products through bond breakages [76] and the formation of active sites, which expedite, for example, oxidation-reduction, oxidation evolution, and hydrogen evolution reactions [77]. The dual application of heteroatom e.g., N-S doped mesoporous carbon was reported by [78]. There is, however, a paucity of literature on the application of iodine (I), boron (B) sulfur (S), lithium (Li), and magnesium (Mg) heteroatoms [79]. The doping process promotes catalytic oxidation, photocatalytic reactions, and catalytic decomposition [80]. Further surface functionalization methods include the production of graphitic structures [31]. However, this process is considered difficult, as it requires higher temperature ranges of ≥ 2800 °C. While some studies have reported on a single-step approach to modifying SC, a combined or dual modification process is proposed as yielding high adsorption abilities [81].

Modification by metal complexes (FeO_2_) was found to significantly enhance SC and increase chemical contaminant adsorption rates in a batch experiment [82]. Similar results by modification with sulfur, chloride, and metal oxides (CuO-ZrO_2_) are reported [83]. Because carbon materials including SC possess evenly distributed internal structures, these elements are uniformly dispersed in the matrix structure at the molecular level promoting effective activation. Nanoparticles’ modification of carbon materials by nano zero-valent iron (nZVI) has been recognized due to their large specific surface area and associated high reactivity towards organic pollutants. nZVI’s established effectiveness in improving the sorption abilities of carbonaceous materials lies in the collection of contaminants in its reactive iron centers [84]. While the iron system is capable of degrading contaminants, distinct reaction mechanisms might differ due to the different sorption properties of the system. Once nZVI is impregnated with AC (nZVI/AC), contaminant adsorption predominantly occurs at the inner AC surface, therefore, separated from ZVI reactive species production sites [85], which is opposite for pure ZVI as both reaction and adsorption take place on the iron surface. Meanwhile, a hindering aspect to its application is related to corrosion effects after long-term applications, which may further generation sources of pollutants in the treatment system [86]. This retards the performance of nZVI in the remediation process. The process of sulfidation of carbo-iron based materials has revealed insights into overcoming this limitation. This is achieved by the addition of sulfur compounds during nZVI synthesis [87]. These surface functionalization methods tend to increase oxygen functional (carboxyl, hydroxyl, and phenolic) groups on the SC surface.

Further modification of SC using biological activation has been explored [21]. Unlike other biological-based carbonaceous materials (e.g., biochar, bone), the impregnation of SC with a microorganism reveals a physical-biological relationship between a non-biological material and a biological material. The modification process enhances surface positive charges and hydrophilicity properties of SC. The high porosity of SC serves as host sites for bacterial action eventually acting as a preferred bio-carrier and promoting biocompatibility. The SC bio-carrier functions by immobilizing microorganisms on its surface while creating a higher biomass concentration, higher metabolic activity, and greater resistance to environmental contaminants. Additionally, since microorganisms possess hydrophilic surfaces, they attract the hydrophilic portions of SC by grafting and by adhesion, contaminant removal efficiencies are enhanced. Further adsorption process may occur if the microorganism surface is negatively charged, hence becoming more adhered to positive charges of SC. Therefore, by increasing positive charges on SC surfaces through chemical addition, biological modification may be highly efficient in the removal of contaminants [88].

## 4. Contaminant Adsorption

Contaminants tend to describe a wide array of chemical, physical, and biological agents that cause deleterious environmental effects. Such contaminants may be either organic or inorganic and generated from either natural sources or anthropogenic sources [38]. Irrespective of their location in environmental media (air, soil, water), their removal or degradation can be challenging since contaminants can transform and exist in different molecular states. The significant properties of SC were discussed in previous sections, and technically, SC is considered a suitable carbonaceous material for engineering adsorbents which can sustain the removal of contaminants. Although there is a lack of research and paucity of data in this area, unlike other carbon materials (e.g., biochar, activated carbons), the value of SC adsorption research may be a focused point in the future for researchers. A relative number of studies have concentrated on the removal/reduction of gaseous pollutants and other volatile organic compounds (VOCs) during secondary re-gasification processes. Since SC is derived from different sources, the surface electrical properties, organic structure, and surface functional groups that are influencing adsorption mechanism factors tend to differ [89]. When supported as a precursor, SC catalyst may promote direct adsorption of contaminants [55].

To efficiently remove or treat contaminants by SC depends on the nature of the contaminant and its properties. It is therefore obvious that while removal mechanisms of some contaminants may require small quantities of SC sorbent application, others may require large quantities of application before reaching sorption removal efficiencies. Variation in the organic structure, surface functional groups, and electrical properties of both SC and contaminants are important determining factors in the adsorption removal process. The presence of surface oxygen functional groups on SC surfaces contribute significantly to the contaminant adsorption process while acidic, amine, and other aromatic groups partly contribute electron-acceptor donor interactions. During SC production processes, charged surface groups are established, which ignite electrostatic attraction, however, depending on the size of each atomic charge present and their relative distances [89]. If negative charges dominate, it is easier to attract positively charged organic compounds thereby quickening the adsorption process. It may therefore be necessary, during the surface functionalization process, to introduce more negative functional groups. The different factors involved in the adsorption mechanism of SC are illustrated in Figure 5.

### 4.1. Gaseous Pollutants

Greenhouse gases and flue gases are generated from natural decomposition and industrial combustion processes. If not captured and treated from source to reduce their pollutant toxicity levels, they tend to cause atmospheric damages through the formation of tertiary pollutants e.g., Ozone (O_3_). Gases such as NO_x_ and SO_2_ are known to cause both environmental and public health issues. Technologies such as wet desulfurization methods are reported as a common means to reducing these gases, however it is coupled with high operating cost and secondary pollution [90]. Application of SC for adsorption and removal of such gases at the source generation is recently increasing in research. Typical of any carbonaceous material, degradation of inter structural formation held by hydrogen, van der Waals forces, hydrophobic interactions, partition uncarbonized fraction, π electrons, and π-π interactions of SC provides a breakthrough for adsorption [38]. The presence and abundance of oxygen-containing functional groups (C═O and C—O), π-π electron-donor acceptor interactions, and phenolic hydroxyl groups on its surface contribute to the process, the number of which differ based on the type of SC. Lignite SC has a high predominance of these formations [34].

The adsorption process may occur first by saturation, however surface formations, e.g., tar and other impurities, cause a lower mass transfer hence initiated resistance to the adsorption. Except for surface functional groups, factors include temperature (kinetics and thermodynamics) and moisture content [62]. Increased reaction temperature results in high micropore formation, which facilities uptake and adsorption of gases. To reduce SC surface resistance for gaseous removal, chemical and biological modification methods have been tested. Increased chemical loading events establish effective surface reaction, which raises the mass transfer, facilitating the uptake and adsorption of pollutant gas. However, high and intense loading events may cause chemical agglomeration, eventually blocking micro and mesopores and limiting pore diffusion of gases unto SC [91]. Graphene oxide (GO), with its characteristic honeycomb carbon atom, _SP_^2^, and _SP_^3^ hybrid orbital structure, has modified SC for further catalytic functions [19]. GO provided an SC surface area of 1072.08 m^2^/g with an iodine adsorption capacity of 1233.99 mg/g. GO has a large number of functional groups, excellent mechanical properties, and chemical stability, which allows it to be used as an efficient carbon material to alter and adjust pore structure, specific surface area, and enhance more developed properties of SC for gas adsorption and storage [92].

The elimination of CH_4_ was investigated by activating SC with methane-oxidizing bacteria (MOB), as SC acts as a successful bio-carrier [21]. Results indicated that CH4 concentration removal was higher (15.02% and 11.11%) than unmodified SC. Further, the MOB SC revealed superior ECs (17.88% and 11.29%) higher than unmodified SC. The MOB was capable of providing a substrate biofilm capable of immobilizing CH_4_ for adsorption and removal. The negative surface charges of the MOB and hydrophilic attraction to the SC defines the adsorption mechanism. The modification processes enhanced the number of oxygen vacancies, which improved the oxidation processes for CH_4_ removal. In other studies, SC modification by Fe^+^, for removal of H_2_S, a harmful gas, was investigated. Surface hydroxyl groups on SC contributed oxygen for oxidation of sulfur with Fe_2_O_3_ formation play a significant role in the oxidation process [36]. By modification of SC with melamine and manganese oxide (MnO), nitrogen oxide (NO_x_) was adsorbed by SC functional groups C=O and N–O, providing sufficient active sites for adsorption. Overall, NO adsorption was highest within 125–200 °C with the highest adsorption at 59.2% at 200 °C [81]. Similar results were obtained [93], however water vapor and SO_2_ generated in the flue gas system may lag the denitrification process by forming a film of water on the SC surface which establishes an isolation layer preventing adsorption. Lignite prepared SC as precursors of desulfurization was found to have a sulfur adsorption capacity of 3.69 g/100 g of the sorbent. The pore structure and the relative changes under high-pressure impregnation supported by oxygen functional groups influence the adsorption process [34]. Similarly, modified SC supported by Fe_2_O_3_ sorbent was employed for the desulfurization test for which the sorbent obtained a 9% yield capacity [94]. The adsorption results of some noxious gases on SC is shown in Table 5. Generally, it is observed that higher adsorption of gases occurred on SC subjected to higher pyrolysis and activation temperature. This further infers the important role of heating regime and temperature control in the adsorption process.

### 4.2. Organic Pollutants

Organic pollutants are ubiquitous in the environment and are mainly generated from natural and anthropogenic sources [97]. The priority list of organic pollutants includes typical pesticides, industrial chemicals, and other unintentional by-products of industrial processes. Because organic pollutants extend their ecological and toxicity effects through food webs, relatively accessible and cost-effective means are being researched to remove them or lower their distribution toxicities, which is a research focus for adsorption. Methods to remediate organic pollutant contamination in soil or water range from bioremediation, stabilization, in-situ chemical oxidation, among others [38]. It is worth noting that the production of SC in itself results in a large amount of wastewater containing more than 300 kinds of organic and inorganic pollutants, therefore a major contributor to organic pollutants in the environment [98]. Despite its carbonaceous formation, the application of SC for organic pollutant adsorption is scant in the literature. The few cited studies have concentrated on its removal effect on organic compounds in water and wastewater with limited application to soil media. Relatively few studies have focused on applying SC for organic contaminant adsorption while some studies have directly applied lignite or its naturally oxidized form as activated coke. The adsorption behavior of some organic pollutants on SC is shown in Table 6.

Industrial wastewater is composed largely of phenolic compounds, which if not improperly handled, causes surface groundwater contamination. These are toxic even at low concentrations and the preferred treatment methods before discharge have been by in-situ chemical oxidation followed by adsorption. For the removal of phenols in groundwater, it was revealed that dispersive forces existing in π-electron and π-π interactions of the aromatic nucleus in the SC and phenol as the main adsorption mechanisms, which resulted in the 35% removal rate [57]. The exterior surfaces of SC are described to be significant in the adsorption process as adsorption rates occur at fast rates. By nZVI (nano zero-valent) modification with SC, 81.3% of coal tar was removed from wastewater and the high percentage removal associated with nZVI proved the necessary high surface activity/active sites, which generated H^+^ atoms inducing C═C, thus breaking the phenol molecular structure; hence, degradation occurred [99]. By this, it is inferred that the modification of SC has significant potential for the removal of organic pollutants. However, phenolic compounds were removed from wastewater within 2 h after the subsequent increase in initial SC dosages signifying that adsorbent concentration and contact time were proportionally related [102]. The effect of initial adsorbate concentrate, adsorbent dosage, and the adsorbent–adsorbate contact time are related factors. Therefore, intraparticle diffusion by pore formation is not the only rate-controlling player, but physio sorption mechanisms may occur. Higher removal rates of benzene and carbon tetrachloride in the gas and aqueous phase were observed for long flame coal and gas coal SC under alkali treatment at 447 mg/g (benzene) and 410 mg/g (carbon tetrachloride) while lean coal SC recorded the least adsorption capacity of 225 mg/g. Similar high adsorption results were obtained in the aqueous phase [107]. These higher adsorption rates are related to the effects of alkali functionalization, which enhances micro-porosity and surface areas.

The use of dyes adds brightness to our aesthetic lifestyles; meanwhile, dye wastewater continues to be a threatening pollutant source as the textile sector increases globally. The high chemical composition of dyes, high chemical stability, high resistance to oxidative, and photodegradation renders even small amounts hazardous to ecosystems. Several studies have reported on the application of low-cost biomass adsorbents for dye removal. There is, however, a paucity of literature on SC treatment of dyeing wastewater. Due to lignite’s low surface capacity, functionalization is necessary to facilitate adsorption processes. Copper-modified lignite was capable of removing 369 mg/g of yellow-brown D3G (DYB) [108]. pH dependence, electrostatic, and chelating interactions, due to additional atoms by Cu, were observed as synergistic factors for dye adsorption. Leonardite, a naturally weathered form of lignite, was demonstrated to efficiently remove congo red dye in aqueous solution after several carbonizations [109]. SC’s effective adsorption of dyes can be associated with the following factors in an effective order of magnitude i.e., dosage of adsorbent > initial concentration > pH > temperature, to achieve a maximum removal yield of 98.97% [100].

Production of lignite-activated coke (LAC) is gaining momentum as an efficient pollutant removal agent in wastewater and other aqueous systems. LAC was earlier reported efficient for toxic gas removal in the treatment system [110]. Characteristic features of LAC to lignite SC were described by [111]. LAC could remove oil from the wastewater mechanism of chemisorption associated with carboxylic, phenolic, and lactonic formations on the surface of the LAC and existing hydrogen bonding interactions [112]. Also, an inverse relationship between adsorbent dosage and pH was observed when an increase in dosage from 2.0 g/L to 4.0 g/L resulted in a slow increase in pH more than a dosage increase from 0 g/L to 2.0 g/L did. A similar inverse relationship was established by [113] as LAC produced at 300 °C showed a high affinity towards the sorption of diclofenac sodium at low concentration than LAC at 700 °C. An exothermic-physisorption process was similar observed by LAC removal of TNT red water [114]. Xyloid lignite (Xylite), which is a commercialized lignite product, was found to be less efficient to remove hydrophilic compounds (68–80%) than hydrophobic compounds. Meanwhile, when mixed with sand, the lignite based material showed high removal efficiencies, which can be associated with interactions with soil microorganisms, which formed a biofilm to facilitate a biosorption process [115].

### 4.3. Heavy Metals

Heavy metals, whether in soil or water, present significant threats to ecological biodiversity and human beings. Their removal from the environment has been studied under various systematic methods including adsorption isotherms [39]. The adsorption removal of heavy metals by carbonaceous materials have been related to pore sizes, surface area, and contact time [39]. In their investigation, [116] discovered more than 99% of Arsenic was removed from groundwater samples by leonardite, oxidized weathered lignite. High adsorption is achieved with smaller particle sizes as they provide large, wider surface-active sites; therefore, carbonization at different temperatures and heating rates with subsequent grounding to obtain fine grains is essential. The authors indicated that particle sizes ≤ 75µm are suitable and efficient. In a comparative study with black coal and other waste coals, it was determined that SC adsorption of Mn, Cr, Fe, Cu, Zn, and Ni from sulfuric acid (H_2_SO_4_) was better than hydrochloric acid (HCl) [117]. The high pH in sulfuric acid is attributed to the influence of the adsorption process.

At relatively high pH (>9.6), SC surfaces become negatively charged and adsorption is less due to the repulsion of like charges. Since SC contains some amount of heavy metals [118], there is a likelihood of ionic competition for active site bindings, which can delay adsorption processes. Hence, high ionic strength may hinder adsorption by SC. Meanwhile, [119] explains that some metals may have higher ionic radius such as Pb^2+^ and Ni^2+^, which provides for stronger adsorption. According to [120], the discussion on ionic strength has been centered around the molecular size and surface concentration. In terms of mass size, removal of Pb, Cd, Cu, and Zn from aqueous solution by lignite was favored by Pb as it showed a high affinity for uptake [121]. Lignite’s with high humic substances appear to be good adsorbents while lignite’s with high inorganic substances exhibit poor adsorption behavior.

## 5. Adsorption Isotherms

Adsorption isotherms quantify contaminants adsorbed unto the adsorbent surface at equilibrium concentration at a constant temperature [122]. Since the adsorptive properties of SC are naturally significantly low, it becomes of utmost importance to test the most suitable adsorption isotherm to assess the success of its real application for contaminant removal. Various adsorption models developed by two parameters or three parameters have been described [123]. For the removal of heavy metals (Pb, Cd, Zn, Cu) from aqueous solution, adsorption data fitted better to the Langmuir isotherm than the Freundlich isotherm as the coefficients of correlation were (R^2^ = 0.992, 0.956, R^2^ = 0.868, 0.715), respectively [121]. This could generally mean that the maximum heavy-metal adsorption occurs best under monolayer conditions of the adsorbate on the surface of the adsorbent. Similar results were observed by [116], where both Langmuir (R^2^ = 0.9815, 0.997) and Freundlich (R^2^ = 0.9963, 0.9906) isotherms fitted well for As(III) and As(IV) data, respectively; however, As(III) data fitted better to the Freundlich isotherm. The discrepancies in the mechanism could be assigned to competing for anion effects on the surface in aqueous solution leading to switching effects in the process.

The Langmuir–Freundlich isotherm (R_2_ = 0.9949, q_m_ = 225.95 mg g^−^) was revealed as a better adsorption data-fitting model compared to the general Langmuir (R^2^ = 0.8525, q_m_ = 369 mg g^−^) and Freundlich (R^2^ = 0.8709, q_m_ = 369 mg g^−^) isotherms, respectively, in the adsorption of direct yellow-brown dye by Cu modified lignite SC [108]. Similarly, [114] describes a Redlich–Peterson isotherm for adsorption of COD and TNT, which revealed a higher correlation effect (R^2^ = 0.998 at 40 °C, q_m_ = 47.4 mg g^−1^) than traditional Langmuir and Freundlich isotherms. Further, [42] tested the Toth and Liu isotherm models against earlier isotherms tested by [114] and found the Toth isotherm to yield a higher correlation effect (R^2^ = 0.9974, q_m_ = 192.53 mg g^−1^) for adsorption of p-nitrophenol (PNP). The Toth isotherm model displays an adsorption effect based on heterogeneous surfaces, which could infer the adsorption of other organic-based contaminants. However, in a similar comparative study on four phenolic compounds, Sips isotherm (R^2^ = 0.98985) derived from the theory of Langmuir and Freundlich isotherms provided a better data fit compared to the Redlich–Peterson isotherm (R^2^ = 0.96215) and the Toth isotherm (R^2^ = 0.97755), but adsorption data for PNP were better fitted by Toth isotherm as confirmed by earlier [42]. For adsorption of dyes by SC activated carbons, Freundlich isotherms were discovered to fit better (R^2^ = 0.92795, 0.95508) than Langmuir isotherms (R^2^ = 0.74361, 0.93466) for methyl orange and industrial blue dye, respectively [65]. For adsorption and immobilization of adsorbate (Atrazine) in soil, [124], demonstrated performance of six adsorption isotherms and observed their correlation (R^2^) effect in order (Freundlich 0.9917 > Langmuir, 0.994 > Linear, 0.9912 > Jovanovic, 0.9830 > Temkin, 0.9301 > Hill, 0.8177). Results indicated that adsorbate adsorption best fitted Freundlich isotherms with a corresponding increase in adsorbate concentration. This confirms previous results reported by [42] that SC adsorption of organic contaminants may occur at maximum heterogeneous surfaces.

## 6. Reusability and Regeneration

A significant criterion for determining ‘low cost’ sorbents are their potential to be reused or regenerated, hence not requiring a refreshing material after every treatment application process [60]. This aspect of adsorbent technology defines its economic potential. Cost and economic analysis related to SC and direct lignite-formed adsorbents are reported [109]. The adsorbent regeneration process can be described as an inverse process of adsorption, involving two main processes i.e., adsorbate desorption and adsorbate decomposition [125]. The involved reaction mechanism comprises an outward diffusion, inward diffusion, followed by adsorption on the adsorbent solid surface. These processes are temperature, pH, and concentration-dependent [126]. Research on the regeneration of SC sorbents remains scant, however the following methods are cited: Thermal regeneration, ultra-sonic water rinse, ultrasonic ammonium rinsing, and thermal vapor regeneration [127].

### 6.1. Chemical Regeneration

Chemical regeneration SC adsorbents involve washing the adsorbent (e.g., NaOH, acetone) and drying at 100–110 °C to study the composition of formative products at various stages by an established equilibrium between the adsorbent and adsorbate [128]. The recent demonstration indicates that approximately 92.7% of CO_2_ equilibrium initial adsorption capacity by activated SC can be yielded after 10 times of adsorption-desorption cycles [62]. The results were consistent with [129]. After four cycles of desorption-adsorption processes, PNP concentration decreased from 121.33 mg g^−1^ to 96.4 mg g^−1^ yielding approximately 79.5% of the initial adsorption capacity [42]. The reduced equilibrium and loss of capacity may be attributed to incomplete desorption processes of SC as micropore adsorption are too low to facilitate solvent regeneration. Chemical regeneration processes are however found to be better than thermal regeneration and suitable for high concentration and low boiling point organic matter adsorbent [38].

### 6.2. Microwave Irradiation Regeneration

Microwave irradiation regeneration involves utilizing microwave to heat sorbents over multiple cycles, which leads to the uniform distribution of metal oxides on support surfaces of adsorbents. The method increases the electron density of surface atoms and results in a high concentration of surface elements and optimization of pore size formation. MW sorbents are stable and exhibit good regeneration ability. Recent results indicated that activated SC can maintain a 98–99% regeneration capacity within 30 s. Similar high regeneration capacities for SO_2_ were reported after 17 cycles [130]. However, long irradiation power may cause carbon loss leading to slow regeneration rates and eventual decline [101]. A new method of regeneration by microwave-ultraviolet (MW-UV) system is available, which is cited to recover carbon nanotubes (CNTs) at 100% within a time of 2.5 min, maintaining a capacity of 80% even after five cycles [131]. This can be applied to SC regeneration. Comparatively, microwave irradiation yields higher adsorption efficiencies than thermal regeneration [132].

### 6.3. Ultrasound Regeneration

Ultrasound and ultrasonic regeneration methods have been explored for the decontamination of soil and sediments. Research by [133] observed that within the ranges of 40–1000 kHz, adsorption-desorption of phenols on porous carbons improved and the enhancement was facilitated by surface diffusivity. Meanwhile, inconsistencies associated with this method of regeneration under various conditions are reported by [134]. The stronger the power and intensity of ultrasound, the higher the desorption rates.

### 6.4. Thermal Regeneration

The adoption of thermal regeneration is almost the most common industrial resource recovery mechanism. However, the high heating rates and temperature involved in the process renders its mechanism to yield lower capacities, in most cases less than 80% of initial adsorptive capacities [135]. The process denatures the carbon structures, hence the loss of particularly micropores and decomposition of surface oxygen groups [136].

### 6.5. Biological Regeneration

The process involves stimulating spent porous carbon with micro-organism to rejuvenate regeneration adsorptive capacities. Prior mechanisms involve established a low gradient of desorption as adsorbates with organics are dissolved in an aqueous solution followed by microbial action [137]. The carbon pore structures provide and serve as hosts for microorganisms providing necessary conditions for surface biofilm formation. Intraparticle diffusion into micropores is prominent here as microorganisms secrete enzymes (exoenzymes). By this, biodegradation processes eventually occur over time, and fractions of pollutants in contact with the carbon get degraded [138]. The merits of this approach come with its low cost and ready availability as microorganisms abound in soil or wastewater.

## 7. Conclusions

Rapid development and industrialization will continuously present several environmental pollution incidences. Finding opportunities for remediation of already polluted environmental media (soil and land) and preventing the excessive damage of new areas is the new norm for environmental management. A traditional environmental engineering approach has been the reuse of by-products and solid waste from agricultural, manufacturing, and industrial operations as precursors to developing remediation materials. Semi-coke from low-rank coal is gaining popularity as a low-cost contaminant/pollutant adsorbent due to its significant physicochemical properties. The research application on semi-coke has mainly focused on its reutilization in the power/energy sector due to its high carbon content. Nevertheless, available research on its application for contaminant removal/adsorption is scanty with the relative studies focused on a few organic pollutants, heavy metals, and noxious gas emissions. The structural complexation of semi-coke is greatly influenced by temperature while surface behavior and chemistry are affected by pH and the addition of other chemical activators. Other factors, including electrostatic interactions, π-π electron donor–acceptor relationships, hydrogen bonding, hydrophobic interactions, and intraparticle diffusion are a few that enhance semi-coke’s capacity for contaminant removal. As a material with high capacity for reusability and regeneration, it is essential to establish an economic output over conventional sorbents. Positive results of semi-coke’s regeneration were observed. This review reiterates the science of semi-coke as a low-cost sorbent material with significant adsorption application for contaminant removal.

## Figures and Tables

**Figure 1 materials-13-04334-f001:**
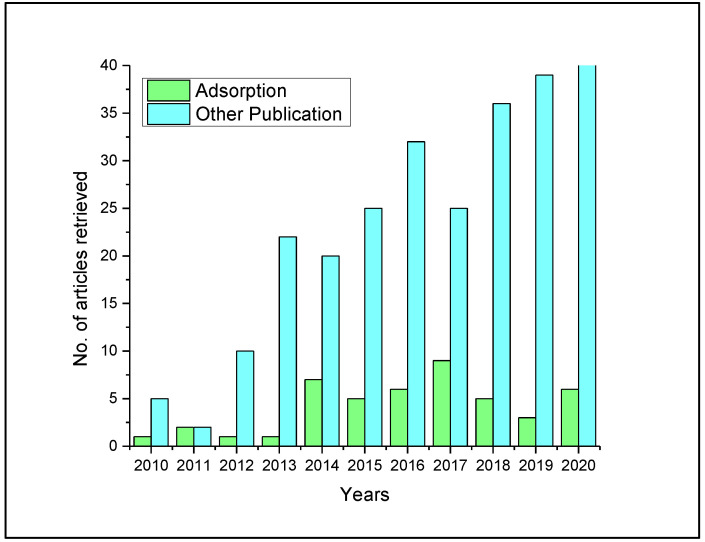
Recent research progress on semicoke adsorption.

**Figure 2 materials-13-04334-f002:**
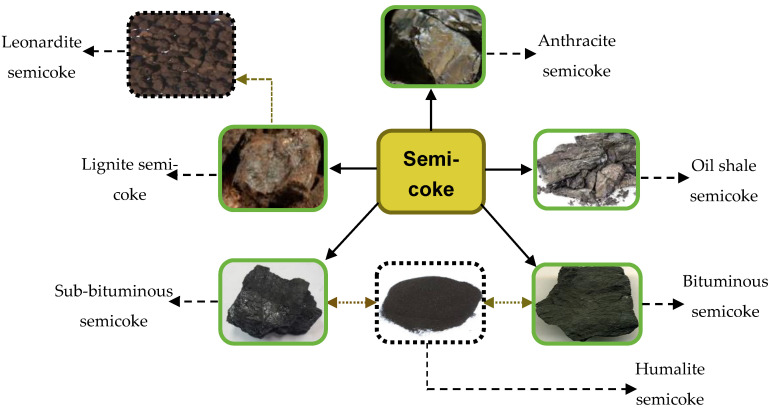
Classification of semicoke types after processing.

**Figure 3 materials-13-04334-f003:**
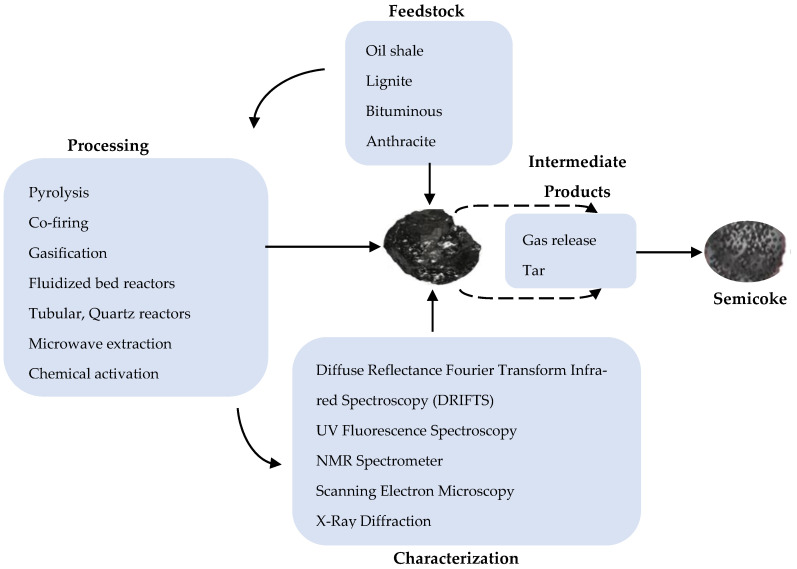
Flow process of some semi coke preparation factors.

**Figure 4 materials-13-04334-f004:**
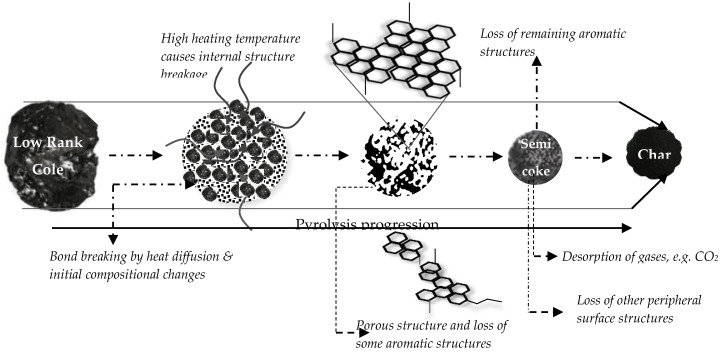
Illustrative process formation of semicoke.

**Figure 5 materials-13-04334-f005:**
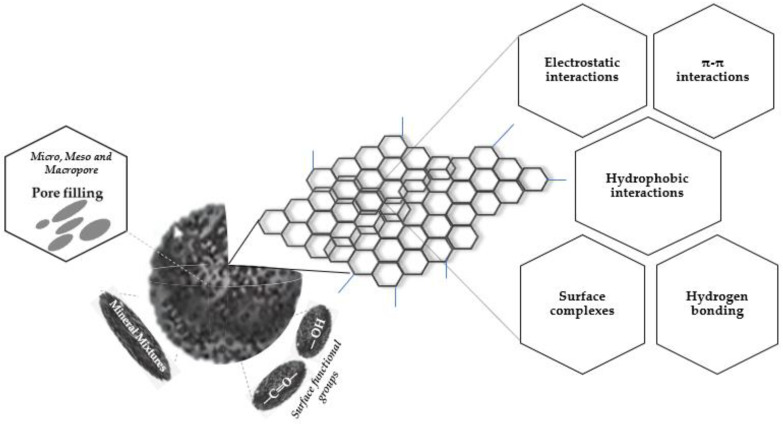
Involving factors of SC adsorption mechanism.

**Table 1 materials-13-04334-t001:** Categorization of adsorbents.

Category	Description	Example
1.	Natural material	Sawdust, wood, earth, bauxite
2.	Natural materials treated to develop their structures and properties	Activated carbons, activated alumina, silica gel
3.	Manufactured materials	Polymetric resins, zeolites, aluminosilicates
4.	Agricultural solid waste and industrial by-products	Date pits, fly ash, red mud
5.	Bio-sorbents	Chitosan, fungi, bacterial biomass

**Table 2 materials-13-04334-t002:** Characteristics of some semi-coke.

Type/Source of Semi-Coke	Specific Characteristics	Reference
Oil shale SC	High ash content, low heat value, High phenol, polycyclic aromatic hydrocarbon (PAHs)	[32,33]
Lignite SC	Large porosity & roughness, higher pyrolysis reactivity, high water absorption, high heating value, small specific surface area, less dense internal strength, high volatile matter, high ash content	[34,35,36]
Bituminous coal SC	High volatility, low viscosity, low ash, low sulfur, low aluminum, high fixed carbon, high specific surface resistance	[37]

**Table 3 materials-13-04334-t003:** Some temperature and activation influenced changes in the SC surface area and pore volume.

Type of SC	Temperature Range (°C/K)	Modification Method	Activation Agents	SSA (m^2^g^−1^)	PV (cm^3^g^−1^)	Reference
Lignite	723 K	-	-	1.03	14.1	[11]
	1023 K	-	-	41.4	2.91	
Oil shale	500 °C	-	-	12.1	0.05	[18]
Bituminous coal	Plasma(40 W) and microwave (60 W)	Physical and plasma	-	84.9	0.03	[58]
		Chemical	HNO_3_, KOH, H_2_O_2_			
	Plasma (40 W), H_2_O_2,_ and microwave (60W)	physicochemical	KOH			
Lignite	700 °C	Chemical	CeO	45.71	2.83	[59]
Lignite	600 °C	Chemical	TEPA + HCL	15.2	0.031	[60]
	800 °C			25.2	0.046	
Lignite	109 °C	-	-	211.9	0.18	[61]
		Chemical	HCl	315.5	0.34	
Lignite	700 °C	Chemical	NaOH	4.7	0.01	[62]
		Chemical	HNO_3_	2.8	0.007	
		Physical	N_2_	20.5	0.04	
-	700 °C	Chemical	Fe (NO_3_)_3_ + Co (NO_3_) + SiO_2_	266.4	0.32	[63]
			Fe (NO)_3_ + Co (NO)_3_	272.1	0.51	
Lignite	120 °C	-	-	247.5	0.29	[64]
	500 °C	-	Fe(NO_3_)_3_.9H_2_O	470.6	0.39	

(SSA-specific surface area, PV-pore volume).

**Table 4 materials-13-04334-t004:** Elemental characteristics of some SC.

Type of SC	Ultimate Analysis (wt%)					Proximate Analysis (wt%)				Reference
	C	H	N	S	O	FC	A	V	M	
Shengli lignite	81.13	5.23	12.00	1.14	0.50	46.24	13.18	40.68	-	[9]
Shanxi lignite SC	73.74	5.05	0.83	0.39	16.60	60.36	3.39	36.25	-	[11]
Urumqi lignite SC	64.95	6.64	0.87	0.34	19.62	45.59	7.58	46.83	-	
Longku oil shale SC	21.36	1.74	6.13	0.72	0.50	12.83	65.75	17.62	3.80	[18]
Longku oil shale SC	78.33		5.33	1.41	14.93	-	-	-	-	[21]
Zhaotong lignite SC	65.94	4.63	1.54	0.70	27.19	-	16.24	53.08	16.74	[34]
Shanxi SC	66.10	1.18	0.79	0.35	4.48	-	9.00	9.96	18.1	[55]
Zhaotong lignite SC	40.40	0.15	0.77	1.03	1.85	40.00	49.31	4.20	6.49	[64]
Tongda SC	83.98	-	-	-	-	-	7.55	8.47	-	[65]
Subbutiminous Shenhua SC	64.82	64.82	0.79	0.44	3.96	54.20	5.09	29.67	11.04	[66]

(FC-fixed carbon, A-Ash, V-Volatile matter, M-Moisture).

**Table 5 materials-13-04334-t005:** Adsorption capacity of some noxious gases on SC.

Noxious Gas	SC Pyrolysis Temp (°C)	Activation Temp (°C)	Adsorption Mechanism	Adsorption Capacity (q_m_)		Reference
				mg g^−1^	mmol g^−1^	
SO_2_	-	-	-	108		[9]
SO_2_ and NO_x_	800	-	Physisorption/Chemisorption	0.62 (SO_2_) 20.14 (NO_x_)	-	[63]
CO_2_	500	600 700 800	Chemisorption	-	2.81 3.5 2.9	[66]
CO_2_	500	700	-	-	2.68	[68]
SO_2_	400	700 800 900	Physical adsorption	-	33.7 36.6 31.5	[95]
H_2_S	350–550	450 500 550	-	-	5.2 7.0 6.3	[96]

(CO_2_-Carbon dioxide, NOx-Nitrogen oxides, SO_2_-Sulphur dioxide, H_2_S-Hydrogen sulfide).

**Table 6 materials-13-04334-t006:** Adsorption capacity of some organic compounds on SC.

Contaminant/Pollutant	Pyrolysis Temp (°C)	Adsorption Mechanism	% Removal	Adsorption capacity (q_m_) (mg g^−1^)	Reference
Phenol	400	Pore filling, π electrons & π-π dispersion interaction	-	-	[57]
MB	-	Pore filling physio-sorption, chemio-sorption	-	4.287	[58]
Phenol	400	Strong surface complexation, electrostatic interactions	95.88	-	[99]
CR	-	physio-sorption	98.7	-	[100]
MO	-	-	98–99	-	[101]
Phenol and PNP	600	Surface complexation	-	42.75 54.77	[102]
CR, MB, acid Fuchsin, MO		Chemio-sorption, π-π dispersion interaction		526.32	[103]
Phenol, MB	800	-	86.58	Phenol (280) MB (121)	[104]
MB	-	-	-	862	[105]
Oilfield wastewater	550	Pore filling	75.6	-	[106]

(MB-Methylene Blue, MO-Methylene Orange, CR-Congo Red, PNP-P-*nitrophenol*).

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
