# Peer review of "Remediation with Semicoke-Preparation, Characterization, and Adsorption Application"

_materials, 2020, doi:10.3390/ma13194334_

Round 1
Reviewer 1 Report
In this paper, the authors describe an intriguing result dealing with the information on SC, its source and production, adsorption mechanism and removal of polluting contaminants.
However, some improvement in this manuscript is required as below.
- Please specify adsorption isotherm results of Semicoke.
- Please revise the figure graphics as the text in the graphic is not legible/will not be legible in publication (ex. Figures 5, 6).
Author Response
Dear Reviewer,
Attached are responses to comments on the manuscript
Regards

Reviewer 2 Report
The article is a review about semi-coke source evaluation and using in contaminant remediation application by contaminant adsorptions. The work is well structured and is based on a comprehensive bibliography. The presented information is important for the field of coal industry, about the semi-coke more specified.
Beside of these positive features, the manuscript is poor of concrete data and technical relevant information from cited bibliography. For moment, the text is more like a well-documented report with literature sources of data. More experimental results (with concrete values of parameters and trends) will change this perspective. A well written example is Chapter 5.1.
Additional suggestions and comments:
- Abstract:
- is too general, is more like an introduction;
- please explain “LRC” also here as “low-rank coal”.
- Introduction:
- line 33: “include:” instead “include;” (the punctuation problems are also in other parts, e.g. line 91);
- Figure 1 has no explanation in text. Even it is in Introduction I think that it belongs to Chapter 2. Please provide the signification of “Other Publication”.
- Chapter 2:
- line 101: Attention, appears “Error! Reference source not found” which is not clear what it is (appears in many places of the manuscript).
- Chapter 3:
- Table 2 is not explained in text. Please provide more details.
- Chapter 4:
-A more detailed explanation for Figure 6 is needed and/or please state in text paragraphs which are related with this figure.
I congratulate the authors for their work.
Author Response
Dear Reviewer,
Please find attached responses to comments on manuscript
Regards

Reviewer 3 Report
The review named "Remediation with semicoke: preparation, characterization and adsorption application" that has been sent to me, in order to be reviewed, is outstanding, from my point of view.
What it noteworthy to mention/highlight is the high innovative method to identify the lack of information in the filed of SC adsorption research/semicoke adsorption, utilization, pollutant removal, which was the rationale of this ingenious metadata historical perspective.
I strongly recommend this state of the art to be published, since it would be a valuable instrument, am initial point for the study of source and production, adsorption mechanism and removal of polluting contaminants,gathering regeneration methods capable of yielding sustainability in the material reuse.
Author Response
Thank you for the kind comments
Round 2
Reviewer 1 Report
The point is very novel, interesting and scientifically important to readers. The present manuscript is acceptable.
Author Response
Thank you